# A Q-LEARNING APPROACH TO THE LOWEST UNIQUE POSITIVE INTEGER GAME

## ABSTRACT

The Lowest Unique Positive Integer (LUPI) game is a multiplayer game where participants attempt to choose the smallest number that no one else selects. While previous studies model LUPI using Poisson–Nash equilibrium assumptions, our work introduces a novel Q-learning-based approach to achieve equilibrium without the need for specific distribution assumptions, such as Poisson. We demonstrate that our Q-learning model successfully emulates the Nash equilibrium while allowing flexibility in the number of players, providing a more robust and practical solution for real-world applications like real-time bidding (RTB) systems. We compare our model's performance against existing Poisson-based strategies, showcasing improved accuracy and adaptability. Furthermore, we apply our model to the Swedish Limbo lottery data and observe significant deviations from theoretical predictions, highlighting the strength of learning-based approaches in dynamic, real-world scenarios.

## 1 INTRODUCTION

The *Lowest Unique Positive Integer* (LUPI) game is a simple multiplayer game where the participants independently select integers from 1 to $\infty$ simultaneously (or capped by some fixed $K$, depending on the variant played). The objective is to choose the smallest number that no other player has selected, in which case there is a unique winner or no winners at all Simon (2023).

We consider the LUPI game as a *normal-form game*, a notion that has been defined in Nowé et al. (2012). The Nash equilibrium (NE) is a key concept in game theory; it is a state where all players follow strategies that are optimal responses among the other ones. Nash (1950) demonstrated that every normal-form game has at least one Nash equilibrium, which may involve mixed strategies. In other words, once a Nash equilibrium occurs, no player in the game can improve their payoff by unilaterally deviating from the equilibrium strategy profile.

In Östling et al. (2011), playing according to the Poisson equilibrium strategy has been tested against the real data from the Swedish lottery game called *Limbo*, introduced by the government-owned Swedish gambling monopoly Svenska Spel on 29 January 2007. Despite the complexity and counter-intuitive properties of the equilibrium strategy, they find it surprisingly close to the observed data. However, notable deviations from the equilibrium prediction are present, including some behaviourally interesting fine-grained discrepancies.

The LUPI players were unaware of the total number of bets. While they could access information about the current count of bets made during the day, they had to place their bets before the daily game closure, thus lacking certainty about the total number of participating players for that day. In Östling et al. (2011), it is assumed that the number of players follows a Poisson distribution with parameter $n$, where $n$ represents the expected number of players. However, the actual number of players in the Swedish LUPI game varies considerably from day to day, which does not align well with the cross-day variance implied by the Poisson assumption. Nevertheless, the Poisson–Nash equilibrium is likely the only computable equilibrium benchmark. Moreover, under the Poisson assumption, Östling et al. (2011) and Pigolotti et al. (2012) derived a recursive formula for a unique symmetric Nash equilibrium for LUPI.

In our study, we employ Q-learning techniques to reach the Nash equilibrium. Our model offers the flexibility to adjust the number of players arbitrarily. This means that we do not assume *a priori*

any particular distribution such as the Poisson distribution or alike. It is worthwhile mentioning that despite a rather theoretical nature, LUPI games, and their possible strategies may be used as a basis for optimising real-time bidding (RTB) systems in reverse auctions, which are the standard ways for publishers to auction off their ad space to advertisers (see Simon (2023); Zeng et al. (2007); Zhao et al. (2014); Zhou et al. (2015) for more details).

## 2 SWEDISH LUPI LOTTERY GAMES AND RELATED WORKS

In the study conducted by Östling et al. (2011), the LUPI game was thoroughly examined, assuming that the number of players follows a Poisson distribution, thus characterising the game as a Poisson game. For further information on the theory of Poisson games, one can refer to Myerson (1998) and Myerson (2000).

Players in a game have individual reward functions that depend on the actions of other players, making the definition of a desired game outcome often ambiguous. It is unrealistic to expect participants to maximise their payoffs, as achieving this goal simultaneously for all players may not be feasible. A central solution concept in game theory is the Nash equilibrium (NE). In a Nash equilibrium, the players all adopt mutual best replies, meaning that each player chooses the best response to the strategies currently employed by the other players. The following definition of NE is provided in (Nowé et al., 2012, Definition 14.3):

**Definition 2.1** *A strategy profile $\sigma = (\sigma_1, \ldots, \sigma_n)$ is termed a Nash equilibrium if, for each player $k$, the strategy $\sigma_k$ constitutes the best response to the strategies $\sigma_{-k}$ of the other players.*

Hence, when a Nash equilibrium is in play, no participant in the game can enhance their payoff by unilaterally altering their equilibrium strategy profile. Consequently, no player has an incentive to change their strategy, and a simultaneous strategy change by multiple players would be required to disrupt the Nash equilibrium.

Myerson (1998) demonstrated the existence of an equilibrium in all games involving population uncertainty with finite action and type spaces, which encompasses Poisson games.

It was demonstrated in Östling et al. (2011) how to derive the formula for the Nash Equilibrium specifically for the Poisson LUPI Game when one allows for a varying number of players, as opposed to the fixed number, as this assumption simplifies the computation of the equilibrium if the number of players is Poisson-distributed. The authors suggested that the Poisson–Nash equilibrium is likely the *only* computable equilibrium benchmark. The authors also explored the real-world version of the LUPI game, the Limbo game, posing the question: do the Limbo players adhere to the Poisson–Nash equilibrium benchmark? To investigate this, they assumed that the number of players is Poisson distributed with a mean equal to the empirical daily average number of choices (53,783). This assumption, however, is flawed due to the actual variation in the number of bets across different days being significantly greater than what the Poisson distribution would predict.

The authors compared the Poisson equilibrium with the field data and found the equilibrium surprisingly close, given its complexity and counter-intuitive nature. However, significant deviations from the equilibrium prediction were evident. Moreover, the authors asked whether an alternative theory could account for both the surprising accuracy of the equilibrium prediction and the systematic deviations observed. To explore this, they posited that different players engage in varying levels of iterated strategic thinking within a cognitive hierarchy (CH).

They stressed that the aim of the cognitive hierarchy model was not merely to better fit the data than the Poisson–Nash model but also to demonstrate how individuals with limited computational capacity might initially approach and eventually converge to such a complex equilibrium. However, they noted that the cognitive hierarchy model only provides indicative evidence of this convergence and should, therefore, be considered an initial step towards a more formal investigation using a learning model.

Theoretical and behavioural analyses suggest reasons why the 'incorrect' theory (Poisson–Nash) might closely mirror actual behaviour in practical scenarios, despite significant empirical discrepancies in the variance of $n$. One rationale is that in straightforward cases (where it is possible to compute the Nash equilibrium for a fixed number of players without resorting to a Poisson dis-

tribution), the equilibria for zero variance (fixed $n$) and for Poisson variance are nearly identical. However, this observation, based on a limited set of examples, may not be universally applicable.

Their second point highlights that field data from Sunday and Monday sessions, characterised by a lower $n$ and reduced standard deviation, show choices very similar to those on other days, despite the variance in $n$ being roughly double. This finding is noteworthy. Nonetheless, given the larger variance in player numbers in the field data compared to that assumed in the Poisson–Nash equilibrium, the Poisson–Nash model can only serve as an approximation.

## 3  GAME THEORY AND MULTI-AGENT REINFORCEMENT LEARNING

For generalities related to Q-learning we refer the reader to Sutton (2018). The central idea of game theory is to model strategic interactions as a game between a set of players. A game is a mathematical object, which describes the consequences of interactions between player strategies in terms of individual payoffs. Different representations for a game are possible. We shall focus on the so-called *normal-form games*, in which game players simultaneously select an individual action to perform. In line with (Nowé et al., 2012, Definition 14.1) we introduce the following definition.

**Definition 3.1** *A* normal-form game *is a tuple* $(n, A_{1,...,n}, R_{1,...,n})$, *where*

- $1, \ldots, n$ *is a collection of participants in the game, called players;*

- $A_k$ *is the individual (finite) set of actions available to player $k$;*

- $R_k : A_1 \times \ldots \times A_n \to \mathbb{R}$ *is the individual reward function of player $k$, specifying the expected payoff he receives for a play* $\mathbf{a} \in A_1 \times \ldots \times A_n$.

A game is played by allowing each player (say, the $k^{\text{th}}$ out of $n$) to independently select an individual action $a$ from its private action set $A_k$. The combination of actions of all players constitutes a joint action or action profile $a$ from the joint action set $\mathbb{A} = A_1 \times \ldots \times A_n$. For each joint action $a \in \mathbb{A}$, $R_k(a)$ denotes $k^{\text{th}}$ agent's expected payoff.

A strategy $\sigma_k \colon A_k \to [0, 1]$ is an element of $\mu(A_k)$, the set of probability distributions over the action set $A_k$ of player $k$. A strategy is called *pure* if $\sigma_k(a) = 1$ for some action $a \in A_k$ and 0 for all other actions, otherwise, it is called a *mixed* strategy. A *strategy profile* $\sigma = (\sigma_1, \ldots, \sigma_n)$ is a vector of strategies, containing one strategy for each player. An important assumption which is made in normal-form games is that the expected payoffs are linear in the player strategies, *i.e.*, the expected reward for player $k$ for a strategy profile $\sigma$ is given by:

$$R_k(\sigma) = \sum_{\mathbf{a} \in \mathbb{A}} \prod_{j=1}^{n} \sigma_j(a_j) R_k(\mathbf{a})$$

with $a_j$ the action for player $j$ in the action profile $\mathbf{a}$.

Depending on the reward functions of the players, a classification of games can be made. When all players share the same reward function, the game is called an identical payoff or common interest game. In the reinforcement learning setting, agents are considered players in a normal-form game, which is played repeatedly to enhance their strategy over time. In a repeated game, all changes in the expected reward are attributed to alterations in the strategy by the players.

There is no changing environment state or state transition function external to the agents. Hence, repeated games are sometimes also referred to as stateless games. Despite this limitation, these games can already present a challenging problem for independent learning agents and are well-suited to test coordination approaches.

Since the expected rewards depend on the strategy of all agents, many multi-agent RL approaches presume that the learner can observe the actions and/or rewards of all participants in the game. This enables the agent to model its opponents and to explicitly learn estimates over joint actions. However, it could be argued that this assumption is unrealistic, as in multi-agent systems which are physically distributed, this information might not be readily available.

As it is generally impossible for all players in a game to maximise their payoff simultaneously, most RL methods aim to achieve Nash equilibrium play. Nash Jr (1950) demonstrated that every normal-form game has at least one Nash equilibrium, possibly in mixed strategies.

## 4 THE Q-LEARNING ALGORITHM SETTINGS FOR THE LUPI GAME

The Q-learning approach applied here adapts the Bellman equation, simplified for the stateless setting. The update rule for Q-values in this case is represented as:

$$Q(a) \leftarrow Q(a) + \alpha[r(t) - Q(a)],$$

where $Q(a)$ is the expected reward for action $a$, $\alpha$ is the learning rate, and $r(t)$ is the immediate reward received from performing action $a$ at time $t$.

We employ Q-learning in the independent setting, *i.e.*, each player maintains an individual vector of estimated Q-values $Q_k(a)$ ($a \in A_k$), where $k$ denotes the player's number. The players learn the Q-values over their own action set and do not utilize any information about other players in the game, except for knowing if no one won (if applicable). Each episode represents another day of the game. The $k$-th player has the same reward function $R_k$, which is 1 when the player won, -1 when they lost, and -0.1 when no one won.

Each player chooses an action based on the current state and the Q-table, using a combination of $\varepsilon$-greedy and softmax strategies. Specifically, with probability $\varepsilon$ (set to 0.95), a random action is chosen (exploration), while with probability $1 - \varepsilon$, an action is selected using the softmax strategy (exploitation). The strategy for $\varepsilon$-greedy can be described as:

$$a = \begin{cases} \text{random action,} & \text{with probability } \varepsilon, \\ \arg\max_a Q(a), & \text{with probability } 1 - \varepsilon. \end{cases}$$

The softmax function used here is a scaled version, where the temperature parameter $T$ (set to 0.15) regulates the degree of randomness in action selection. The probability of selecting action $a$ based on Q-values is given by:

$$P(a) = \frac{\exp(Q(a)/T)}{\sum_{a'} \exp(Q(a')/T)},$$

where a lower temperature $T$ implies less randomness and a higher temperature leads to more diverse action choices.

In our implementation of Q-learning, the following parameters were selected:

- Learning rate ($\alpha$) = 0.01
- Number of episodes (NUM_EPISODES) = 3000

The $\alpha$ parameter controls the learning rate, where a lower value results in slower adjustment of the Q-values, while a higher value may lead to faster but potentially unstable convergence.

## 5 COMPARISON TO NASH EQUILIBRIA

In this section, we analyze the unique symmetric Nash equilibrium for LUPI under the Poisson assumption with an expected player count of $n$. The equilibrium, expressed as $(p_n(1), p_n(2), \ldots)$, was originally identified by Östling et al. (2011) and Pigolotti et al. (2012). The recursive expression for this equilibrium is given by:

$$p_n(1) = \frac{\ln(1 + n)}{n},$$

and for subsequent terms:

$$p_n(k + 1) = p_n(k) + \frac{1}{n} \ln\left(1 - np_n(k)e^{-np_n(k)}\right).$$

We also refer to Srinivasan & Simon (2024) for asymptotics and continuous approximations for these. Below, we present a graphical comparison between this theoretically derived Nash equilibrium and the Nash equilibrium estimated through our Q-learning agent. The comparison visually illustrates how well the Q-learning algorithm approximates the theoretical equilibrium under the Poisson assumption.

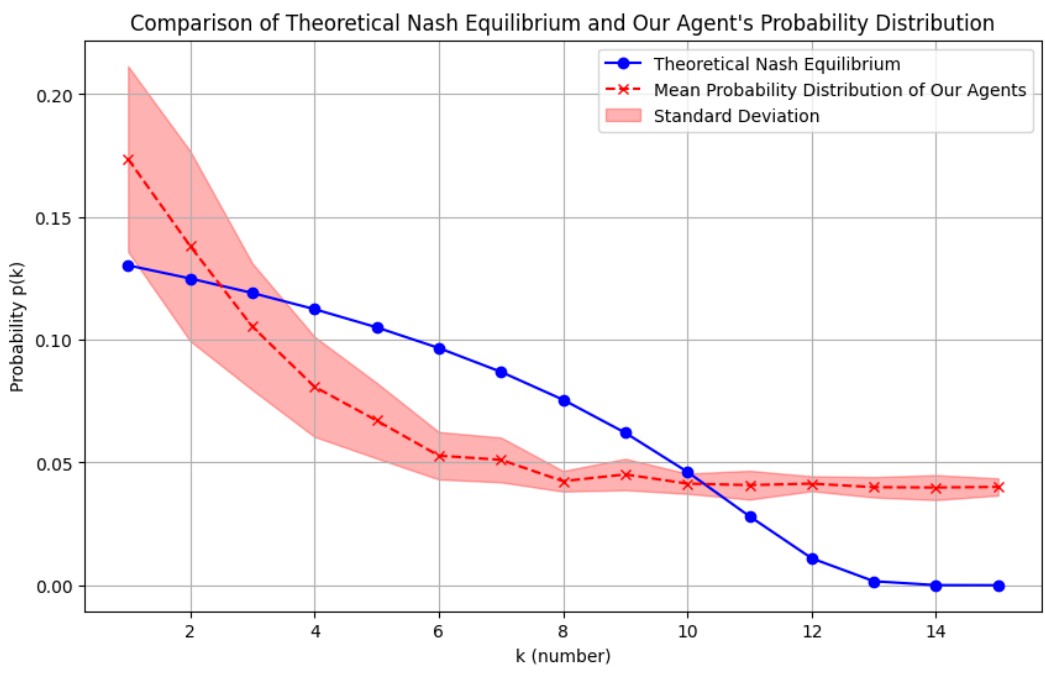

Figure 1

Figure 1 demonstrates the robustness of the Q-learning algorithm in converging to the Nash equilibrium. The minimal discrepancies observed between the theoretical predictions and the empirical results indicate the high accuracy and reliability of the Q-learning method in this context.

## 6  ANALYSIS OF THE LIMBO LOTTERY GAME

We have applied our Q-learning agent to the Limbo lottery game, a Swedish lottery introduced by Svenska Spel in 2007. For our analysis, we first examined 49 days of data from the Limbo game, observing that none of the selected numbers each day were smaller than the winning number. The data used in this analysis are publicly available Östling et al. (2011). To refine our analysis, we excluded the top 700 most popular numbers, leaving a set of approximately 1,000 potentially winning numbers out of the 100,000 possible choices, which implied an estimated 1% chance of winning.

We then tested the performance of our Q-learning agent by incorporating its choices into this dataset. Over 49 rounds (days), we simulated the agent's participation in the game, checking whether it would win under the given conditions. The agent succeeded in winning 8 rounds, which corresponds to a win rate of approximately 16.33%. This result indicates that our agent was able to outperform the expected baseline probability of winning, demonstrating the effectiveness of our approach in real-world lottery scenarios like Limbo.

Here, according to the theoretical distribution, values greater than 20 are seldom chosen. However, we notice that in the actual game, higher values tend to win; see Tables 1–2 for the details.

Table 2: Comparison of predicted agent wins with theoretical outcomes over 49 days.

| Day | Wins | | | Win Indicator | | Players |
|---|---|---|---|---|---|---|
| | Agent Pred. | Actual Wins | Theo. Wins | Agent Win? | Theo. Win? | |
| 1 | 1618 | 7178 | 4 | 0 | 0 | 59993 |
| 2 | 6847 | 5168 | 6 | 0 | 0 | 50446 |
| 3 | 6813 | 5425 | 4 | 0 | 0 | 42226 |
| 4 | 5866 | 5866 | 5 | 1 | 0 | 45508 |
| 5 | 6025 | 5942 | 3 | 0 | 0 | 45928 |
| 6 | 7811 | 7194 | 1 | 0 | 0 | 57126 |
| 7 | 6387 | 6387 | 5 | 1 | 0 | 43363 |
| 8 | 6336 | 5619 | 3 | 0 | 0 | 58431 |
| 9 | 7133 | 5855 | 7 | 0 | 0 | 69030 |
| 10 | 6518 | 6518 | 3 | 1 | 0 | 60144 |
| 11 | 9819 | 6711 | 1 | 0 | 0 | 54920 |
| 12 | 4022 | 6636 | 5 | 0 | 0 | 64725 |
| 13 | 8396 | 5374 | 5 | 0 | 0 | 43203 |
| 14 | 2730 | 2730 | 5 | 1 | 0 | 63040 |
| 15 | 483 | 5103 | 2 | 0 | 0 | 52785 |
| 16 | 6904 | 5844 | 11 | 0 | 0 | 44502 |
| 17 | 1748 | 6899 | 8 | 0 | 0 | 57684 |
| 18 | 365 | 6296 | 1 | 0 | 0 | 59272 |
| 19 | 3805 | 6995 | 1 | 0 | 0 | 58458 |
| 20 | 2216 | 8343 | 5 | 0 | 0 | 45740 |
| 21 | 3949 | 7167 | 1 | 0 | 0 | 49826 |
| 22 | 2291 | 8833 | 3 | 0 | 0 | 41953 |
| 23 | 9052 | 8357 | 7 | 0 | 0 | 48034 |
| 24 | 337 | 8072 | 3 | 0 | 0 | 52159 |
| 25 | 660 | 6619 | 6 | 0 | 0 | 55411 |
| 26 | 431 | 3691 | 7 | 0 | 0 | 40267 |
| 27 | 2611 | 7137 | 3 | 0 | 0 | 62832 |
| 28 | 1857 | 7098 | 3 | 0 | 0 | 38133 |
| 29 | 1690 | 7798 | 3 | 0 | 0 | 58936 |
| 30 | 4768 | 4768 | 6 | 1 | 0 | 54489 |
| 31 | 7923 | 6851 | 3 | 0 | 0 | 40062 |
| 32 | 1901 | 6996 | 10 | 0 | 0 | 64423 |
| 33 | 1992 | 7911 | 1 | 0 | 0 | 54938 |
| 34 | 6082 | 6082 | 2 | 1 | 0 | 47127 |
| 35 | 6327 | 6327 | 10 | 1 | 0 | 40005 |
| 36 | 2673 | 4506 | 5 | 0 | 0 | 61477 |
| 37 | 3678 | 3678 | 2 | 1 | 0 | 55240 |
| 38 | 9880 | 5913 | 3 | 0 | 0 | 54518 |
| 39 | 631 | 5389 | 3 | 0 | 0 | 61259 |
| 40 | 3693 | 6843 | 3 | 0 | 0 | 56371 |
| 41 | 5212 | 5212 | 3 | 1 | 0 | 49496 |
| 42 | 5585 | 5585 | 9 | 1 | 0 | 61404 |
| 43 | 2603 | 6567 | 10 | 0 | 0 | 50985 |
| 44 | 9516 | 6778 | 2 | 0 | 0 | 57265 |
| 45 | 6246 | 6246 | 2 | 1 | 0 | 47350 |
| 46 | 9218 | 7891 | 3 | 0 | 0 | 61156 |
| 47 | 584 | 6436 | 1 | 0 | 0 | 58425 |
| 48 | 9538 | 6633 | 3 | 0 | 0 | 48978 |
| 49 | 4871 | 4871 | 7 | 1 | 0 | 47116 |

Table 1: Summary Statistics

| Statistic | Value |
|---|---|
| Total rounds | 49 |
| Total wins | 8 |
| Win percentage rate | 16.33% |
| Theoretical total wins | 0 |
| Theoretical win percentage rate | 0.00% |
| Average number of participants | 52982 |

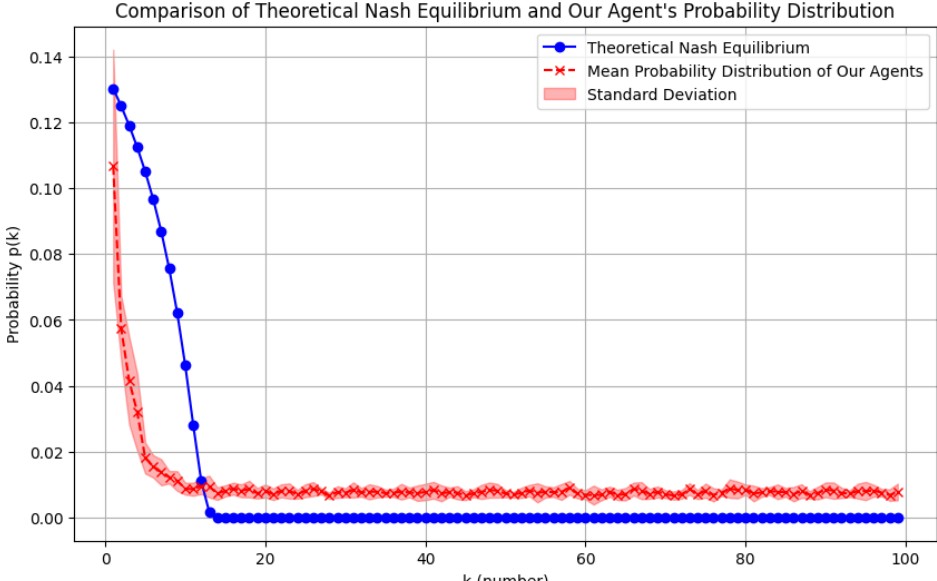

Figure 2: The probability distribution chart for $k = 100$.

Next, we slightly modified the results from the actual game, meaning we limited the maximum selectable number to number=1000. There was no chance of winning, so we set the best choice to a winning one (if there was a winning choice, we did not change it), and we removed the best choices to give a 10% chance of winning. Specifically, considering the results lower than the winning one, we removed 100 numbers with the fewest selections. Then we calculated the average number of players, which was approximately 16,000. We trained the agent for this number of players and checked the theoretical results. The agent won 6 rounds out of 49, while the theoretical 'agent' won nothing. The tables below present these results (see Table 3 and Table 4).

Table 3: Summary Statistics

| Statistic | Value |
|---|---|
| Total rounds | 49 |
| Total wins | 6 |
| Win percentage rate | 12.24% |
| Theoretical total wins | 0 |
| Theoretical win percentage rate | 0.00% |
| Average number of participants | 16608 |

## 6.1 CONCLUSIONS

The presented figure compares the theoretical Nash equilibrium with the probability distribution of our trained agents over multiple game iterations. As shown in Figure 3, the theoretical distribution,

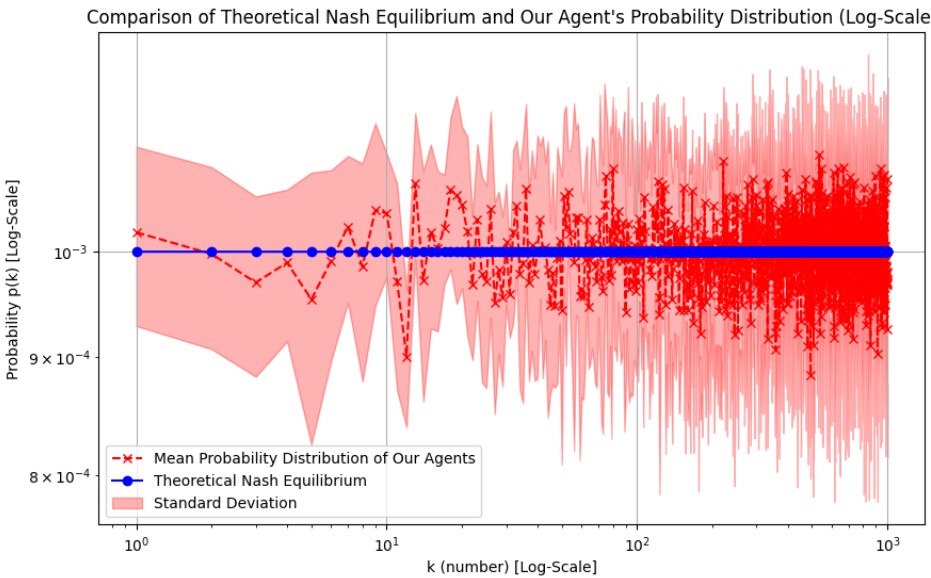

Figure 3: The probability distribution chart for $k = 1000$.

which assumes a Poisson distribution of players, predicts a nearly uniform strategy. However, as evident from the plots above, our agents, trained using Q-learning, deviate from this theoretical prediction, particularly for higher values of $k$, where their probability distribution exhibits significant fluctuations.

The large standard deviation observed for higher $k$ values suggests that our agents explore a wider range of strategies, likely as a result of the Q-learning process optimising for a *fixed* number of participants. This exploration allows agents to discover non-trivial strategies that are not captured by the theoretical model. Despite the noise and variation, our agents' performance, as indicated by the summary statistics, exhibits a superior win rate compared to the theoretical prediction, further supporting the hypothesis that Q-learning agents find more effective strategies in real-world scenarios with fixed $n$ (see Table 3).

Figure 3 highlights a key difference: while the theoretical Nash equilibrium assumes optimal play under Poisson-distributed participants, our agents' learning approach adapts to the fixed number of players, leading to more successful outcomes. The deviation from the theoretical distribution indicates that, in complex environments where theoretical assumptions do not hold, learning-based agents can outperform static equilibrium models.

In conclusion, these findings suggest that the theoretical model, based on unrealistic assumptions, may not be sufficient to capture the true dynamics of the game. In contrast, learning algorithms such as Q-learning, tailored to specific game conditions, provide a more robust and effective method for strategy development, as demonstrated by the higher win rate achieved by our agents.

Table 4: Comparison of predicted agent wins with theoretical outcomes over a 49-day period

| Day | Wins | | | Win Indicator | | Players |
|---|---|---|---|---|---|---|
| | Agent Pred. | Actual Wins | Theo. Wins | Agent Win? | Theo. Win? | |
| 1 | 16 | 536 | 349 | 0 | 0 | 13848 |
| 2 | 409 | 655 | 959 | 0 | 0 | 11956 |
| 3 | 577 | 994 | 641 | 0 | 0 | 17891 |
| 4 | 64 | 772 | 755 | 0 | 0 | 19483 |
| 5 | 359 | 230 | 718 | 0 | 0 | 12588 |
| 6 | 666 | 850 | 322 | 0 | 0 | 12962 |
| 7 | 223 | 402 | 791 | 0 | 0 | 13027 |
| 8 | 309 | 839 | 864 | 0 | 0 | 15806 |
| 9 | 168 | 168 | 19 | 1 | 0 | 12172 |
| 10 | 929 | 970 | 444 | 0 | 0 | 19353 |
| 11 | 490 | 490 | 969 | 1 | 0 | 15388 |
| 12 | 771 | 934 | 469 | 0 | 0 | 33743 |
| 13 | 598 | 344 | 727 | 0 | 0 | 11023 |
| 14 | 954 | 284 | 614 | 0 | 0 | 16063 |
| 15 | 6 | 922 | 201 | 0 | 0 | 35014 |
| 16 | 509 | 284 | 8 | 0 | 0 | 15461 |
| 17 | 844 | 349 | 136 | 0 | 0 | 15474 |
| 18 | 809 | 740 | 518 | 0 | 0 | 22671 |
| 19 | 759 | 251 | 680 | 0 | 0 | 14943 |
| 20 | 89 | 658 | 89 | 0 | 0 | 15575 |
| 21 | 662 | 300 | 761 | 0 | 0 | 20399 |
| 22 | 736 | 794 | 782 | 0 | 0 | 16212 |
| 23 | 686 | 280 | 210 | 0 | 0 | 19311 |
| 24 | 667 | 690 | 482 | 0 | 0 | 11865 |
| 25 | 850 | 695 | 297 | 0 | 0 | 12848 |
| 26 | 300 | 274 | 897 | 0 | 0 | 8644 |
| 27 | 911 | 733 | 95 | 0 | 0 | 13332 |
| 28 | 255 | 445 | 189 | 0 | 0 | 11147 |
| 29 | 125 | 945 | 209 | 0 | 0 | 16385 |
| 30 | 284 | 284 | 346 | 1 | 0 | 14614 |
| 31 | 557 | 162 | 186 | 0 | 0 | 13064 |
| 32 | 366 | 835 | 995 | 0 | 0 | 18501 |
| 33 | 520 | 675 | 499 | 0 | 0 | 26034 |
| 34 | 141 | 141 | 622 | 1 | 0 | 19922 |
| 35 | 698 | 505 | 837 | 0 | 0 | 13501 |
| 36 | 85 | 85 | 315 | 1 | 0 | 19321 |
| 37 | 64 | 906 | 337 | 0 | 0 | 19038 |
| 38 | 850 | 504 | 82 | 0 | 0 | 22818 |
| 39 | 885 | 585 | 120 | 0 | 0 | 18061 |
| 40 | 485 | 485 | 13 | 1 | 0 | 11838 |
| 41 | 695 | 344 | 120 | 0 | 0 | 18644 |
| 42 | 24 | 915 | 77 | 0 | 0 | 18323 |
| 43 | 700 | 564 | 113 | 0 | 0 | 15096 |
| 44 | 725 | 885 | 683 | 0 | 0 | 14256 |
| 45 | 367 | 596 | 891 | 0 | 0 | 13414 |
| 46 | 339 | 206 | 814 | 0 | 0 | 9528 |
| 47 | 965 | 844 | 108 | 0 | 0 | 17589 |
| 48 | 510 | 870 | 167 | 0 | 0 | 14130 |
| 49 | 945 | 660 | 466 | 0 | 0 | 21562 |

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
