# OpenReview forum: "A Q-learning approach to the Lowest Unique Positive Integer game"
_ICLR.cc/2025/Conference — Submitted to ICLR 2025_

### Official Review · Reviewer_yYCy · 2024-10-29

**Soundness:** 1
**Presentation:** 1
**Contribution:** 1
**Rating:** 3
**Confidence:** 4

**Summary:**

This paper studies the Lowest Unique Positive Integer (LUPI) game, where participants independently and simultaneously select integers. If the smallest number is chosen by a single participant, then that participant wins the game, otherwise, there is no winner.

The equilibria of this problem have been studied in previous work under the assumption that the number of participants follows a Poisson distribution. This paper studies the behavior of agents that use a simple Q-learning-based approach to the problem. The authors experimentally prove that the result obtained by the Q-learning-based approach behaves better than the expected payoff at the equilibrium under the Poisson arrival model.

**Strengths:**

The paper is confusing and the writing is poor. I struggle to find the strengths of this paper.

**Weaknesses:**

This paper is clearly below the ICLR bar. The writing is highly suboptimal, the various paragraphs do not have any clear relationship. It is not clear what the technical contribution of the paper is.
At best, the authors have implemented a simple off-the-shelf algorithm and evaluated it on a specific dataset, comparing it against a Nash Equilibrium computed under specific assumptions (i.e., Poisson Arrival).

**Questions:**

No question

---

### Official Review · Reviewer_LBap · 2024-10-31

**Soundness:** 3
**Presentation:** 1
**Contribution:** 2
**Rating:** 3
**Confidence:** 2

**Summary:**

This paper studies a multiplayer game, LUPI, in which the agents aim to choose the smallest number that no one else chooses. The authors provide a Q-learning-based approach that independently learns the Q function of each agent and provides experiment results to show the effectiveness of their proposed algorithm. Unlike previous works, they do not need some assumptions on the number of players, like the Poisson assumption, making it more applicable in the real world.

**Strengths:**

1. The idea of using a Q-learning-based approach for the classical LUPI game is novel in the literature.

2. The experiment shows that the proposed algorithm performs well and can explore and discover some non-trivial strategies beyond the theoretical prediction.

3. The motivation is clear: Since the Poisson assumption does not generally hold in reality, a robust approach that can hold without prior knowledge of the number of players is useful in reality.

**Weaknesses:**

1. The writing can be improved. Since the main focus of the paper is LUPI, it is necessary to give more introduction to LUPI, like some general notations and the mathematical definitions and equations to make it more friendly to the reader. For example, instead of only referring to previous papers, it would be better to introduce some notations like $p_n(k)$. Also, since $n$ represents the number of players and $k$ is the selected number,  should the captain of Figure 2 be $K=100$, where $K$ is the maximum value of the selected number?

2. Overall, this paper appears to apply independent Q-learning directly to a classic multi-player game, which may seem like a straightforward application of the independent Q-learning method. Could you highlight any unique modifications or theoretical insights that go beyond the direct application of existing algorithms?

3. The authors claim that Figure 1 shows how well the Q-learning algorithm approximates the theoretical equilibrium under the Poisson assumption.  However, in Figure 1, the two lines don’t appear to exhibit a strong similarity. Could you clarify this further?

4. In Section 6, the author states that "To refine our analysis, we excluded the top 700 most popular numbers, leaving a set of approximately 1000 potentially winning numbers out of the 100000 possible choices". Could you elaborate on the motivation behind excluding these numbers and discuss what effect it might have if they were included?

5. There is no comparison of this work and some previous algorithms. Is there any previous practical algorithm for the LUPI game? It would be better to compare this paper with theirs. Including a comparative analysis would strengthen the paper, as the baseline "which implied an estimated 1% chance of winning" and the theoretical 0% win rate may seem too weak.

6. Would it be possible for the authors to derive some theoretical analysis to better illustrate why the higher values tend to win in the actual game, but values greater than 20 are seldom chosen theoretically? I understand that the rebuttal time is limited. Some intuitive explanations will also be appreciated.

**Questions:**

The questions are contained in the weakness part.

---

### Official Review · Reviewer_Vy8V · 2024-11-04

**Soundness:** 2
**Presentation:** 1
**Contribution:** 1
**Rating:** 3
**Confidence:** 4

**Summary:**

The authors apply Q-learning to obtain novel experimental insights on a Swedish lottery game called Limbo over classical theoretical understanding of the LUPI game based on Poisson distributional assumptions.

**Strengths:**

The authors obtain novel insights over prior theoretical work using the Poisson-Nash assumption by &Ouml;stling et al. by applying Q-learning to real world data.

**Weaknesses:**

Most of the paper is focused on basic literature review. Q-learning is an extremely well-known algorithm. The novel portion of this paper is the application of this well-known algorithm to the LUPI setting. Furthermore, since the theoretical contribution is not significant, it would be nice to see application of Q-learning some other real-world LUPI games besides Swedish Limbo.

Additionally, the paper is not written well. Sections of the paper feel disjointed. For example, it is odd that in Section 2, the authors start making definitions on things like NE and explaining basic game theory (it seems like this belongs in Section 3). Additionally, for example, Table 2 appears before Table 1.

While the approach leads to some interesting empirical results (at least so it seems, as mentioned above and in the questions below, some restructuring seems required to make the paper and its results more readable), the contributions are significant enough for publication at ICLR.

**Questions:**

1. I would suggest not reusing captions for tables; it is not informative to have repeated captions because it defeats the point of the caption and doesn't allow disambiguation between the tables.
2. It is a bit odd in the first line of the second paragraph that you using the wording "normal-form game, a notion that hs been defined in Nowe et al." The unnecessary verbosity combined the wording, seems to suggest somehow that normal-form games were first defined by Nowe et al. I don't think a reference is required for this, and this entire phrase after the comma can be dropped.
3. It seems you used the wrong template: there should be line numbers during review, and the header says 2024 not 2025.

---

### Meta-Review · Area_Chair_9hWP · 2024-12-20

**Metareview:**

This paper received pretty low scores and the authors did not write any rebuttal. I guess they decide to quiet-withdraw this paper.

Reject

**Additional Comments On Reviewer Discussion:**

No rebuttal, no discussion

---

### Decision · Program_Chairs · 2025-01-22

Reject